# Multi-Omics Analysis of Gut Microbiota and Host Transcriptomics Reveal Dysregulated Immune Response and Metabolism in Young Adults with Irritable Bowel Syndrome

**DOI:** 10.3390/ijms25063514

**Published:** 2024-03-20

**Authors:** Jie Chen, Tingting Zhao, Hongfei Li, Wanli Xu, Kendra Maas, Vijender Singh, Ming-Hui Chen, Susan G. Dorsey, Angela R. Starkweather, Xiaomei S. Cong

**Affiliations:** 1College of Nursing, Florida State University, Tallahassee, FL 32306, USA; 2School of Nursing, Yale University, Orange, CT 06477, USA; t.zhao@yale.edu; 3Department of Statistics, University of Connecticut, Storrs, CT 06269, USA; hongfei.2.li@uconn.edu (H.L.); ming-hui.chen@uconn.edu (M.-H.C.); 4School of Nursing, University of Connecticut, Storrs, CT 06269, USA; wanli.xu@uconn.edu; 5Microbial Analysis, Resources, and Services (MARS), University of Connecticut, Storrs, CT 06269, USA; kendra.maas@uconn.edu; 6Computational Biology Core, Institute for Systems Genomics, University of Connecticut, Storrs, CT 06269, USA; vijender.singh@uconn.edu; 7Pain and Translational Symptom Science, University of Maryland School of Nursing, Baltimore, MD 21201, USA; sdorsey@umaryland.edu; 8College of Nursing, University of Florida, Gainesville, FL 32610, USA; a.starkweather@ufl.edu

**Keywords:** irritable bowel syndrome, gut microbiome, gene expression profiling, immunity, metabolic processes, multi-omics

## Abstract

The integrated dysbiosis of gut microbiota and altered host transcriptomics in irritable bowel syndrome (IBS) is yet to be known. This study investigated the associations among gut microbiota and host transcriptomics in young adults with IBS. Stool and peripheral blood samples from 20 IBS subjects and 21 healthy controls (HCs) collected at the baseline visit of an RCT were sequenced to depict the gut microbiota and transcriptomic profiles, respectively. The diversities, composition, and predicted metabolic pathways of gut microbiota significantly differed between IBS subjects and HCs. Nine genera were significantly abundant in IBS stool samples, including *Akkermansia*, *Blautia*, *Coprococcus*, *Granulicatella*, *Holdemania*, *Oribacterium*, *Oscillospira*, *Parabacteroides*, and *Sutterella*. There were 2264 DEGs found between IBS subjects and HCs; 768 were upregulated, and 1496 were downregulated in IBS participants compared with HCs. The enriched gene ontology included the immune system process and immune response. The pathway of antigen processing and presentation (hsa04612) in gut microbiota was also significantly different in the RNA-seq data. *Akkermansia*, *Blautia*, *Holdemania*, and *Sutterella* were significantly correlated with *ANXA2P2* (upregulated, positive correlations), *PCSK1N* (downregulated, negative correlations), and *GLTPD2* (downregulated, negative correlations). This study identified the dysregulated immune response and metabolism in IBS participants revealed by the altered gut microbiota and transcriptomic profiles.

## 1. Introduction

Irritable bowel syndrome (IBS) is a complex and multifactorial disorder manifesting abdominal pain and symptoms including anxiety, depression, fatigue, etc., and its exact underlying mechanisms are not fully understood [1,2,3]. A combination of genetic, environmental, and microbial factors are likely to influence IBS. Accumulated evidence suggests the gut–brain axis, which involves bidirectional communication between the gut and the central nervous system, plays a crucial role in IBS [4,5,6]. Moreover, the specific genes implicated may differ among individuals and across different subtypes of IBS. Advancements in high-throughput sequencing technologies and comprehensive multi-omics approaches contribute to a more comprehensive understanding of gene expression in IBS [7,8]. 

The interaction of the gut–brain axis and transcriptomics has been confirmed to mediate the pathophysiology of IBS [7,8]. Recent studies also demonstrate that gut microbiota and related metabolism mediate IBS relapses [7,9]. Our recent study reported the distinguishing distribution of gut microbiota compositions between young adults with IBS and healthy controls and the effect of diet on gut microbiota diversity and compositions [10,11]. Research on the role of gut microbiota in young adults with IBS has been ongoing, and the results have, indeed, been inconsistent. Previous studies found that dysregulated immune responses, including elevated pro-inflammation and inflammation, and altered lipidome metabolism, are related to the development and progression of IBS [12,13,14,15]. However, specific genes related to immune response and metabolism in young adults with IBS were not universally agreed upon across studies. 

Exploring the relationship between gut microbiota and host transcriptomics can provide insights into the molecular and genetic factors contributing to the development and progression of IBS. Investigating the interaction between gut microbiota and host gene expressions involved in young adults with IBS is crucial for advancing our understanding of the etiologic mechanisms of the disorder, developing personalized treatment strategies, identifying biomarkers, and ultimately improving patient outcomes. In this study, we focus on multi-omics data using the gut microbiome 16S rRNA sequencing data and transcriptomic data from a randomized control trial (RCT, NCT03332537) [16]. This multi-omics data analysis aimed to (1) depict the different diversities, composition, and predicted metabolic pathways of gut microbiota between young adults with and without IBS; (2) describe transcriptomic profiles including differentially expressed genes (DEGs), differentially enriched gene ontology, different pathways between young adults with and without IBS; and (3) explore the association among those significantly enriched genera, downregulated and upregulated genes, as well as the associated functions.

## 2. Results

### 2.1. Phenotypic Characteristics of Subjects

The majority of IBS subjects and HCs were non-Hispanic, White female, and college students. The detailed characteristics are summarized in Table 1. IBS subjects reported significantly higher pain severity (worst pain and least pain in the last 24 h, average pain, and current pain), pain interference, anxiety, and fatigue than the HCs (Table 2). IBS subjects also reported higher levels of depressive symptoms and sleep disturbance (Table 2). 

### 2.2. Differentiations of Gut Microbiota between IBS Subjects and HCs

All the 20 IBS subjects had valid gut microbiota data. Among the 21 HCs, one did not mail back the stool sample, and one was removed from the final analysis due to the low quality of sequencing reads, yielding a total of 39 stool samples included in this analysis. The IBS group demonstrated a significantly lower coverage, higher Invsimpson index, higher Shannon index, and higher total observed species (sobs) (Figure 1(a1–a4)). The principal coordinate analysis (PCoA) plot (Figure 1b) of the Bray–Curtis dissimilarity index showed that the IBS subjects and HCs were partially overlapped. The LEfSe results indicate that there are nine genera significantly enriched in IBS, including *Parabacteroides*, *Akkermansia*, *Blautia*, *Coprococcus*, *Granulicatella*, *Holdemania*, *Oribacterium*, *Oscillospira*, *Parabacteroides*, and *Sutterella* (Figure 1c). A compositional relative abundance for the 39 subjects was plotted on average according to the nine genera (Figure 1d). Furthermore, there were 34 significantly different predicted KEGG pathways between the IBS participants and HCs, including the enriched pathway “Antigen processing and presentation” in HCs and the enriched pathway “Butanoate metabolism” and “Phosphatidylinositol signaling system” in IBS subjects (Figure 1e).

### 2.3. Differentiated Expressed Genes (DEGs) between IBS Subjects and HCs

Using the Deseq 2 analysis, we identified 2264 DEGs between the IBS participants and HCs, with 768 upregulated and 1496 downregulated in IBS subjects (Figure 2a). The principal component analysis (PCA) plot of gene expression profiling showed no overlap between IBS participants and HCs (Figure 2b). Gene ontology analysis indicates that the GOs among the top 50 upregulated genes comprise the GO 004419 biological process involved in interspecies interaction between organisms, GO 0050896 response to stimulus, and GO0002376 immune system process (Figure 2c). Meanwhile, the GOs among the top 50 downregulated genes included the GO0002376 immune system process, GO 0022610 biological adhesion, and GO 0050789 regulation of biological process (Figure 2d). By uploading the identified 2264 DEGs into Qiagen IPA analysis, we found that these DEGs were involved in several disease pathways, e.g., antigen presentation pathway and interferon signaling (Figure 2e). These GOs and disease pathways among the identified DGEs highlight the dysregulated immune response among the host transcriptomics in IBS subjects compared with the HCs.

### 2.4. Associations of the Differentiated Gut Microbiota and Gene Expressions 

We found 170 significant correlations of the nine genera (horizontal axis) and the top 50 upregulated and 50 downregulated DEGs (vertical axis) by using Spearman correlation as presented in Figure 3. Among the top 50 upregulated genes, *Sutterella* was significantly correlated with 21 DEGs, *Akkermansia* was significantly correlated with 20 DEGs, and *Blautia* and *Holdemania* were significantly correlated with 18 DEGs. Regarding the top 50 downregulated genes, *Akkermansia* was significantly correlated with 18 DEGs, *Blautia* was significantly correlated with 14 DEGs, *Sutterella* was significantly correlated with 11 DEGs, and *Holdemania* was significantly correlated with 8 DEGs. Among the nine genera and the upregulated DEGs, *RPL9P33* was significantly correlated with seven genera, *ANXA2P2* was significantly correlated with six genera, and *RP11-730G20.2* was significantly correlated with five genera. Among the downregulated DEGs, *PCSK1N* was significantly correlated with five genera, and *GLTPD2* was significantly correlated with four genera.

Figure 4 illustrates some of the significant correlations, including the positive associations between *Blautia* and *DAZAP2P1* (rho = 0.458, *q* = 0.042, Figure 4a), *Blautia* and *ANXA2P2* (rho = 0.482, *q* = 0.042, Figure 4b); and the negative associations between *Akkermansia* and *FTH1P2* (rho = −0.499, *q* = 0.042, Figure 4c), and *Sutterella* and *ADAT3* (rho = −0.393, *q* = 0.043, Figure 4d). In addition to the significant correlation between taxa and genes, we found that several abundant metabolic pathways in IBS gut microbiota were significantly enriched in genes, such as the hsa04612, antigen processing and presentation (Figure 5). Figure 5a shows the enriched hsa04612 in IBS stool samples, and Figure 5b shows the enriched hsa04612 in IBS gene expression profiles. The correlation and consistent differential pathway of hsa04612 antigen processing and presentation reveal the dysregulated immune response in young adults with IBS both in the stool and blood samples.

### 2.5. Association of Self-Reported Pain/Symptoms, and the Differential Gut Microbiota and Gene Expressions

Given the different compositions of gut microbiota and gene expressions between IBS subjects and HCs, we further explored if these significantly enriched nine genera, and top 50 upregulated and downregulated DEGs were involved in pain and symptoms among IBS subjects. The results of Spearman correlations (Figure 6 and Figure 7) indicate that no significant correlations emerged between the enriched nine genera and self-reported pain and symptoms, nor between the top 50 upregulated and downregulated DEGs and self-reported pain and symptoms.

## 3. Discussion

By assaying the gut microbiota and whole transcriptomics, we identified the different diversities, compositions, predicted functions of gut microbiota, and the differently expressed genes, gene ontology, and metabolic pathways of these DEGs from a group of young adults with IBS and healthy controls. The results confirm previous findings that IBS subjects experience significant pain severity and pain interferences in daily life, as well as higher levels of anxiety and depressive symptoms [17,18]. These results are also in line with our previous study, which reports the pain and symptom experiences of the whole IBS group (N = 80) [16,17]. Thus, sampling representation in the study was warranted, although the IBS subjects were older than the HCs. We only reported the different genera between the IBS subjects and HCs to homogenize the microbiota data for the multi-omics data analysis. The significantly enriched *Akkermansia*, *Blautia*, *Oscillospira*, and *Parabacteroides* were consistent with our previous findings by comparing the whole group of 80 IBS subjects with the 21 HCs, further validating the current analysis [10].

This study reported the link between gut microbiota and transcriptomic profiles among young adults with IBS, including the specific taxa at the genus level and the association of genera and genes. Both the gut microbiota and DEGs analysis have identified that the pathway “antigen presenting process” was significantly different between IBS subjects and HCs, indicating a contrasting expression of antigen presentation in the DEGs and the gut microbiota (Figure 5), revealing a dysregulated immune response in IBS, and supporting the physiopathology of IBS. We also found that the predicted pathways “butanoate metabolism” and “phosphatidylinositol signaling system” were enriched in IBS subjects’ stool samples (Figure 1d). Both the dysregulated “butanoate metabolism” and “phosphatidylinositol signaling system” had been implicated in gastrointestinal inflammatory states [19,20,21]. A previous DEGs study found that the butanoate metabolism was downregulated in ulcerative colitis [22]. These contrasting expressions of “Butanoate metabolism” in the gut microbiota and DEGs may depict a dynamic interaction of the genera and gene expressions in IBS. Further studies are needed to verify the contrasted enriched pathways in the DEGs and the gut microbiota of IBS subjects. The cohesively dysregulated immune response and metabolism revealed by the gut microbiota and gene expressions implicate potential therapeutic strategies.

Among the nine identified genera, *Akkermansia*, *Blautia*, *Holdemania*, *Parabacteroides*, and *Sutterella* were significantly correlated with upregulated and downregulated genes. Ours and other studies have confirmed the significantly elevated abundance of *Akkermansia*, *Blautia*, *Oscillospira*, and *Parabacteroides* in the intestinal tract of ulcerative colitis and patients with IBS [10,23,24,25]. This analysis highlights the role of *Holdemania* and *Sutterella* in IBS. Previous studies confirm the role of *Sutterella* in small intestine edema and in the development and progress of IBD [26,27]. A multicenter data analysis with 1522 amplicon samples confirms that *Holdemania* is enriched among IBS subjects [28].

Both *RPL9P33* and *RP11-730G20.2* were significantly upregulated in IBS subjects compared with HCs. *RPL9P33* was positively associated with the abundance of *Akkermansia*, *Blautia*, *Coprococcus*, *Granulicatella*, *Holdemania*, *Parabacteroides*, and *Sutterella*. *RP11-730G20.2* was significantly correlated with five genera (*Blautia*, *Oribacterium*, *Oscillospira*, *Parabacteroides*, and *Sutterella*) in our study. *RPL9P33* (https://www.ncbi.nlm.nih.gov/gene/100271229, accessed on 18 January 2024) and *RP11-730G20.2* (https://dice-database.org/genes/RP11-730G20.2, accessed on 18 January 2024) are pseudogene, *RP11-730G20.2* is indexed in the Database of Immune Cell Expression, Expression quantitative trait loci (eQTLs), and Epigenomics (DICE) and significantly enriched in CD4 T cells. The detailed functions of *RPL9P33* and *RP11-730G20.2* are yet to be understood fully. The strong correlations between *RPL9P33*, *RP11-730G20.2*, and specific genera may reveal the underlying function of *RPL9P33* and *RP11-730G20.2* in IBS development through meditating the specific microbiota. Further studies are warranted to confirm these novel findings.

The *ANXA2P2* was significantly upregulated in IBS subjects compared with HCs and positively associated with the abundance of *Akkermansia*, *Blautia*, *Granulicatella*, *Holdemania*, *Oscillospira*, and *Sutterella*. *ANXA2P2* is a potential immunological and prognostic signature gene, impacting the pathogenies and prognosis of ovarian serous cystadenocarcinoma and hepatocellular carcinoma [29,30], and functioning in the progression of glioblastoma [31]. In addition, *ANXA2* acts as a potential biomarker for immune infiltration and prognosis of cancer [32], as well as a potential immunosuppression marker in glioma [33]. These may suggest that *ANXA2P2* is involved in the dysregulated immune response of the development and progression of IBS through entangling the altered gut microbiota, although we recruited emerging adults with IBS experiencing a mild level of pain and symptoms. 

*DAZAP2P1* was significantly upregulated in IBS and is anticipated to facilitate various functions, including the stimulation of lysophospholipids and arachidonic acid release [34]. Arachidonic acid undergoes further metabolism, resulting in the production of certain inflammation mediators. Consequently, pharmacological agents such as phospholipase A2 inhibitors hold significant value for treating inflammatory conditions [35]. Earlier research has indicated the potential of phospholipase A2 inhibitors as anti-inflammatory agents [36]. Additional investigations are necessary to validate the elusive effects of *DAZAP2P1* in the pathobiology of IBS and to explore the potential therapeutic applications of phospholipase A2 inhibitors in managing IBS.

*PCSK1N* was significantly downregulated in IBS and negatively associated with the abundance of *Akkermansia*, *Blautia*, *Coprococcus*, *Parabacteroides*, and *Sutterella*. *PCSK1N* is a shared gene with a typical endocrine expression profile in enteroendocrine cells from the healthy stomach and duodenum [37]. The downregulation of *PCSK1N* among IBS patients in our study indicates abnormal hormone-secretion-related gene expression, which may explain the abnormalities in releasing hormones and irregular bowel movements among IBS patients [38]. This finding requires further intestinal histological or hormonal studies to confirm, since the expression signatures may vary from enteroendocrine cells to peripheral blood samples. On the other hand, the PCSK1 gene exhibits more prevalent variants that have been linked to changes in body mass index, elevated levels of circulating proinsulin, and disruptions in glucose homeostasis, which may also contribute to gastrointestinal disorders, including IBS [39].

The downregulated *GLTPD2* was negatively associated with the abundance of *Akkermansia*, *Blautia*, *Holdemania*, and *Sutterella. GLTPD2* has been reported as a genetic determinant of the human lipidome and in the regulation of inflammation and cell deaths [40]. The findings are consistent with a previous study that demonstrates that elevated lipid levels and pro-inflammatory symptoms characterize colon mucosa of IBS patients [14]. The downregulated *GLTPD2* and the associated genera may also reveal the underlying mechanisms of the intolerance of fatty food among IBS patients [15].

Although we measured patients’ self-reported pain and symptoms, we did not detect a significant association between the different genera, different expressed genes, and self-reported pain and symptoms among young adults with IBS, possibly due to the small sample size. One of the possible reasons could be the high-dimensional characteristics of the gut microbiota and RNA sequence data. In addition, the mechanisms of pain and symptoms in IBS are yet to be illustrated, even though previous studies indicate that the trajectories of IBS pain and symptoms are related to the altered gut microbiota [41,42]. Another possible reason could be that we recruited IBS subjects with a younger age (mean = 21 years old) and experienced a mild level of pain and unpleasant symptoms. Enrolling a group of IBS patients experiencing severe pain and symptoms or longitudinally following up with IBS patients to map the pain and symptoms trajectory may further capture the underlying multi-omics mechanisms of IBS pain and symptoms. 

This study shows the feasibility of multi-omics investigations to explore the underlying mechanisms of symptomology in young adults with IBS, as well as in patients with other conditions. The identified genera and gene markers may reveal targeted intervention, particularly the altered immune response and metabolism that emerged from the gut microbiota and host transcriptomics, including the significantly enriched genera and DEGs and the correlations between these enriched genera and DEGs.

### Strengths and Limitations

To the best of our knowledge, this study is one of the first to explore the involvement of gut microbiota and gene expression changes, as well as self-reported pain and symptoms among young adults with IBS. The differences between subjective measures (self-reported pain and symptoms) and objective measures (gut microbiota and gene expressions) provided new insight into the molecular and genetic factors contributing to the development and progression of IBS and the mechanisms of IBS pain and symptoms. Generalization of findings from this study should be cautious since several limitations need to be addressed. We only included a small sample size of IBS subjects and HCs recruited in one center. However, the gene ontology analysis supported the idea that the IBS group had altered gene expressions by the up- and down-expressed genes related to immune response. Given that the RNA samples of IBS participants and HCs were sequenced in different batches, the batch effect may skew the associations of gut microbiota and differentially expressed genes. However, we adjusted the batch effects by applying the ComBat-seq to minimize the variation in expression data attributed to batch effects. 

## 4. Materials and Methods

### 4.1. Design

This multi-omics data analysis included baseline data from 20 subjects with IBS and 21 healthy controls (HCs) from an RCT (NCT03332537) [16]. Young adults with IBS and healthy control were recruited from October 2016 to March 2019 in the northeastern region of the United States. The RCT protocol was approved by the University Institute Review Board (IRB). Informed consent was acquired from each participant. Privacy was maintained during the entire data collection and management process. Deidentified data were used in the current analysis.

### 4.2. Inclusion and Exclusion of Subjects

The inclusion and exclusion criteria were reported in our previous protocol and trial [16]. Subjects were enrolled in the study if they were (1) 18–29 years of age; (2) able to read and speak in English; (3) able to access the internet; (4) having a clinical diagnosis of IBS according to the Rome III or IV criteria from a healthcare provider (for IBS subjects only). Subjects were excluded if they had (1) chronic pain conditions (e.g., chronic pelvic pain, headache, back pain, etc.); (2) severe mental health conditions; (3) celiac disease or inflammatory bowel disease; (4) infectious diseases; (5) diabetes mellitus; (6) injury or open skin lesions on the non-dominant arm; (7) history of substance abuse or regular opioid use; (8) history of prebiotics/probiotics or antibiotics use in the past months.

### 4.3. Measurements of Variables

#### 4.3.1. Demographic Characteristics and Self-Reported Pain and Symptoms 

Sociodemographic data were collected including sex, age, race, ethnicity, education, and employment status. Self-reported pain and symptoms (anxiety, depression, fatigue, and sleep disturbance) were measured using the Brief Pain Inventory short-form (BPI-sf) and the NIH “Patient-Reported Outcomes Measurement Information System” (PROMIS) in the Research Electronic Data Capture (REDCap). The BPI-sf pain severity included 4 items in pain severity (worst and least pain in the last 24 h, average pain, and current pain) and 7 items in pain interference (e.g., mood, walking ability, etc.). The four domains of RPOMIS anxiety, depression, fatigue, and sleep disturbance were measured in this study. The scoring systems of the BPI and PROMIS were reported in our previous reports [16,18]. A higher score of BPI and T-score of PROMIS indicate a higher level of pain severity/interferences and symptoms, respectively.

#### 4.3.2. Stool Sample Collection and Gut Microbiota Data Sequencing 

Stool samples were collected at home by the participants following our previous protocol using the OMNIgene^®^-gut (DNA Genotek, Ottawa, ON, Canada) kit [11,43]. Training on collection of the samples at home was provided to the participants by research personnel. After collection, participants were instructed to mail the samples back to the research laboratory within a week of the baseline visit data collection using a pre-labeled and pre-paid envelope with a tracking barcode. Once received by the laboratory, the stool samples were aliquoted into small tubes and stored in a −80 °C freezer for bulk processing. A total of 0.25 g stool from each participant was placed into a beat tube and sent out for DNA extraction and sequencing. The 16S rRNA V4 region amplicon was sequenced by Illumina Miseq 2000 platform (Illumina, San Diego, CA, USA) at the University of Connecticut Microbial Analysis, Resource, and Services laboratory following our previous protocol for gut microbiota data analysis [43].

#### 4.3.3. Blood Sample Collection and Whole Transcriptomics Data Sequencing 

Non-fasting peripheral blood samples were collected using PaxGene tubes (Qiagen, Germantown, MD, USA) following the manufacturer’s instructions. Venipuncture was performed by trained registered nurses in the certificated bio-behavior laboratory. Blood samples were stored homogenized in the Paxgene tubes according to the manufacturer’s protocol. The Paxgene tubes were transferred to a −80 °C freezer in the laboratory at our institute until RNA extraction procedures. RNA extraction, quality checks, quantitation, library preparation, and sequencing were performed by the Center for Genome Innovation, part of the Institute for Systems Genomics at the University of Connecticut. Libraries were prepared with the Illumina Nextera Library Preparation kits (Illumina, San Diego, CA, USA). Samples were sequenced on the Illumina NovaSeq 6000 platform using a 150 bp paired-end run (2 × 150). We used FastQC (http://www.bioinformatics.babraham.ac.uk/projects/fastqc/, accessed on 13 May 2023) and Trimmomatic v.0.39 [44] tools for evaluating the quality of the raw sequence data following our previous process [45].

### 4.4. Data Analysis

R version 4.2.0 was used for the data analysis. The demographic characteristics, self-reported pain, and symptoms were summarized with frequency and percentage for categorical variables and the mean and SD for continuous variables. The differences in the demographic characteristics between IBS subjects and HCs were examined using a chi-square and Fisher’s exact tests. The differences in age, self-reported pain, and symptoms between the IBS participants and HCs were investigated using the Wilcoxon rank-sum test.

The raw 16S rRNA sequencing data were processed by the Mothur 1.43.0 software following the analysis pipeline of Miseq (http://www.mothur.org/wiki/MiSeq_SOP, accessed on 13 March 2023) to obtain the taxonomy and diversity of the gut microbiota. Paired-end sequences were combined into contigs and aligned against to the SILVA 132 V4 16 S rRNA gene reference alignment database with poor-quality sequences removed. Operational taxonomic units (OTUs) were determined at a 97% identity [46,47]. Linear discriminant analysis (LDA) effect size (LEfSe) [48] was used to identify the significantly enriched abundant taxa between the microbiota composition of IBS subjects and HCs. A taxonomic cladogram was used to visualize the discriminative features based on the settings including the alpha value of 0.05 for the factorial Kruskal–Wallis test and for the pairwise Wilcoxon test, and the logarithmic linear discriminant score of 2.0 for the LEfSe following our previous analysis [46]. The metabolic functions (KEGG pathways) of gut microbiota compositions were predicted using phylogenetic investigation of communities by reconstruction of unobserved states (PICRUSt) [49]. Function (KEGG pathways at level 3) differences between IBS subjects and HCs were detected using statistical analysis of metagenomic profiles (STAMP) [50], and the Benjamini–Hochberg procedure was used to decrease the false discovery rate for multiple test corrections. The *p*-value was set as 0.05.

The raw blood RNA sequences were aligned to the reference genome (GRCh38.p14/hg38) downloaded from Ensemble 109 [51] using the HISAT2 (http://daehwankimlab.github.io/hisat2/, accessed on 13 May 2023) [52] software. Gene expression was quantified with HTSeq v.2.0.4 [53], a method that regards a gene as the combination of all its exons, irrespective of isoforms, while excluding reads that span multiple genes. To tackle the issue of batch effects, we utilized the software ComBat-seq (https://github.com/zhangyuqing/ComBat-seq, accessed on 13 May 2023) [54], an extension of ComBat [55], a widely employed tool for batch effect correction. Consequently, the batch-adjusted counts, the differentially expressed genes (DEGs) between the IBS and HC groups were assessed by using the R package “DESeq2” (https://www.bioconductor.org/packages/release/bioc/html/DESeq2.html, accessed on 13 May 2023) [56]. Genes were considered differentially expressed if their adjusted *p*-value, estimated using the Benjamini–Hochberg test, was equal to or lower than 0.05. Furthermore, we prioritized the DEGs whose log2 fold change (LFC) absolute value was higher than 1.5. We evaluated these gene sets for enrichment in biological processes and pathway enrichment, including Gene Ontology (GO) [57] and the Kyoto Encyclopedia of Genes and Genomes (KEGG) [58]. We further imported the DEGs into QIAGEN ingenuity pathway analysis (IPA) [59] to find the regulators of the DEGs, as well as the function of these DEGs.

The Spearman correlation was calculated to assess the associations among the differentially enriched genera in stool samples, the top 50 upregulated and downregulated DEGs between the IBS and HC subjects, and self-reported pain and symptoms. The Benjamini–Hochberg method was applied to control the false discovery rate (FDR) at 0.05 for multiple comparisons [60], leading to the calculation of q-values and enabling statistical inference for the correlations. The correlations and significance findings were visualized by heatmap. 

## 5. Conclusions

This study reveals the linkage of gut microbiota and transcriptomic profiles among young adults with IBS and the dysregulated immune response and metabolism in the gut microbiota and transcriptomic profiles from a multi-omics approach. Further study could explore the bi-direction between the alternative gut microbiota and transcriptomic among patients with IBS and develop precise intervention by targeting the altered gut microbiota and host transcriptomics-related molecular and genetic factors and dysregulated immune response and metabolism. 

## Figures and Tables

**Figure 1 ijms-25-03514-f001:**
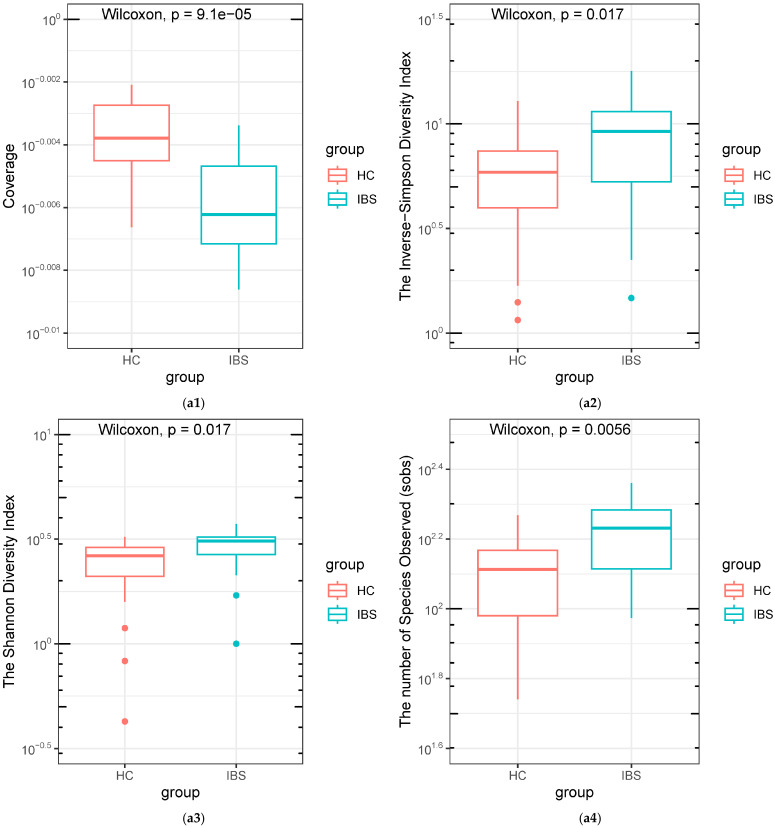
Differentiation of gut microbiota between IBS subjects and HCs. (**a1**–**a4**). Differentiation of coverage, Invsimpson index, Shannon index, total observed species (sobs) between IBS and HC groups. (**b**). PCoA plot of the Bray–Curtis dissimilarity index showed that the IBS subjects and HCs were partially overlapped. (**c**). Compositional difference of gut microbiota between IBS participants and HCs by the LEfSe analysis. The LEfSe results indicate that there are nine genera significantly enriched in IBS, including *Parabacteroides*, *Akkermansia*, *Blautia*, *Coprococcus*, *Granulicatella*, *Holdemania*, *Oribacterium*, *Oscillospira*, *Parabacteroides*, and *Sutterella.* The alpha value for the factorial Kruskal–Wallis test was set to 0.05 for the pairwise Wilcoxon test, and the threshold on the logarithmic LDA score for discriminative features was set as 2.0. Green indicates significantly enriched features in IBS, red indicates significantly enriched features in HCs. (**d**). Compositional relative abundance for the IBS and HC subjects plotted on an average basis according to the nine genera. (**e**). Different metabolic profiles of gut microbiota between IBS subjects and HCs (predicted KEGG pathways).

**Figure 2 ijms-25-03514-f002:**
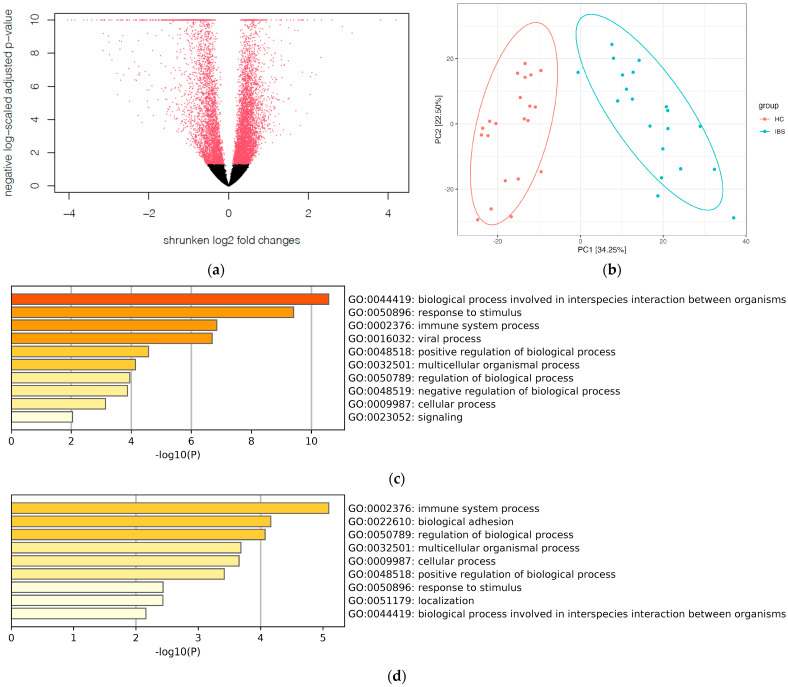
Differentiated expressed genes (DEGs) between IBS subjects and HCs. (**a**). Volcano plot of the DEGs, 2264 DEGs between the IBS subjects and HCs with 768 upregulated and 1496 downregulated in IBS subjects. The black dots indicate genes present in both groups with a difference of log2 fold change (LFC) less than 1.5; The red dots indicate genes present in both groups, but with a difference of LFC equal to or greater than 1.5. (**b**). Principal component analysis (PCA) plot showed no overlap between IBS participants and HCs. (**c**). Gene ontology among the top 50 upregulated genes in IBS participants. (**d**). Gene ontology among the top 50 downregulated genes in IBS subjects. (**e**). Interferon signaling pathway from DEGs. Gene, Protein, and Chemical identifiers marked with an asterisk indicate multiple identifiers in the dataset file map to a single gene and/or chemical in the Global Molecular Network. (**f**). IPA Prediction legend to help in the understanding of pathway from DEGs.

**Figure 3 ijms-25-03514-f003:**
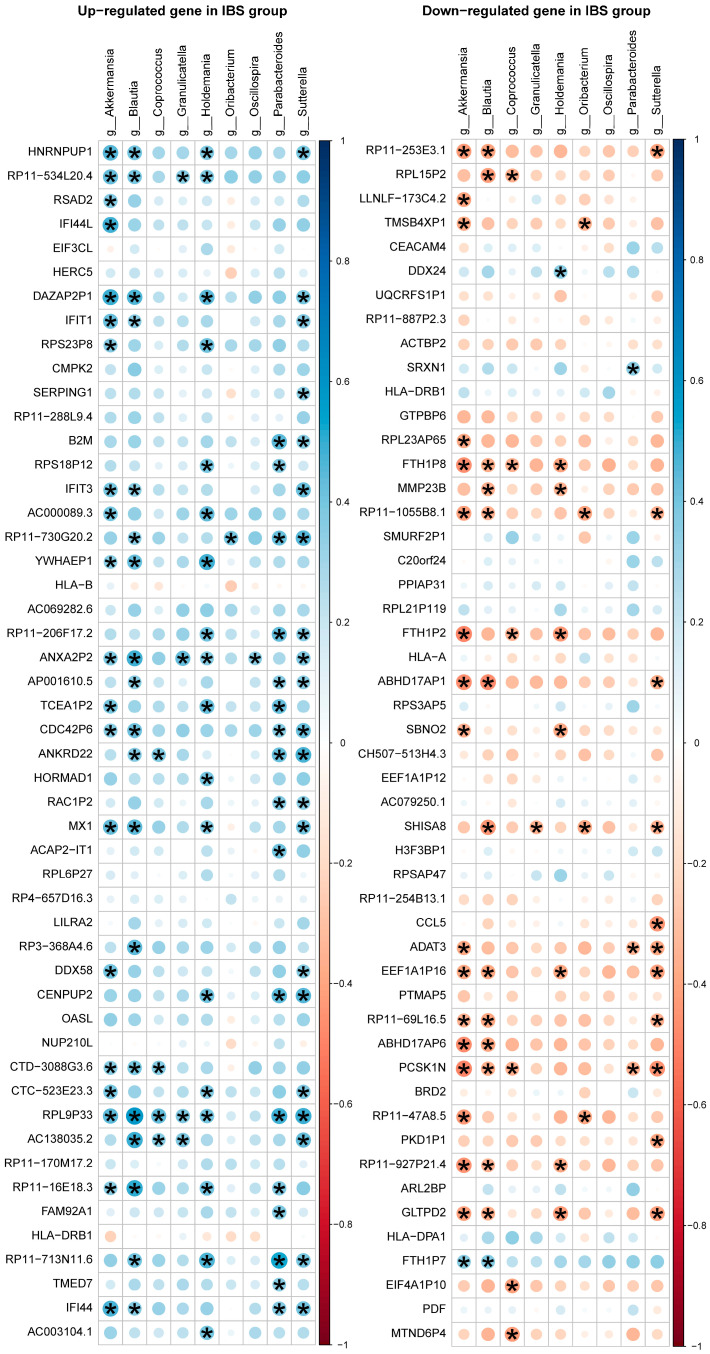
Spearman’s correlation coefficient of gut microbiota and expressed genes. In the left column, the correlations between the nine genera (horizontal axis) and the top 50 upregulated DEGs (vertical axis). In the right column, the correlations between the nine genera (horizontal axis) and the top 50 downregulated DEGs (vertical axis). The significant correlations (*q*-value < 0.05), after multiplicity adjustment by Benjamini-Hochberg method to control the false discovery rate (FDR) at 0.05, were marked with asterisks. Blue indicates a positive correlation, red indicates a negative correlation.

**Figure 4 ijms-25-03514-f004:**
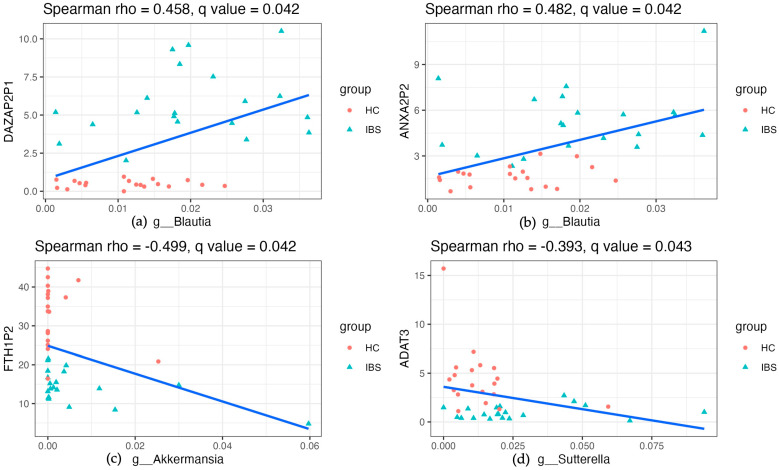
Spearman’s correlation coefficient of the identified genera and genes. The Benjamini–Hochberg method was applied to control the false discovery rate (FDR) at 0.05 for multiple comparisons. (**a**). The positive associations between *Blautia* and *DAZAP2P1* (rho = 0.458, *q* = 0.042). (**b**). The positive associations between *Blautia* and *ANXA2P2* (rho = 0.482, *q* = 0.042). (**c**). The negative associations between *Akkermansia* and *FTH1P2* (rho = −0.499, *q* = 0.042). (**d**). The negative associations between *Sutterella* and *ADAT3* (rho = −0.393, *q* = 0.043).

**Figure 5 ijms-25-03514-f005:**
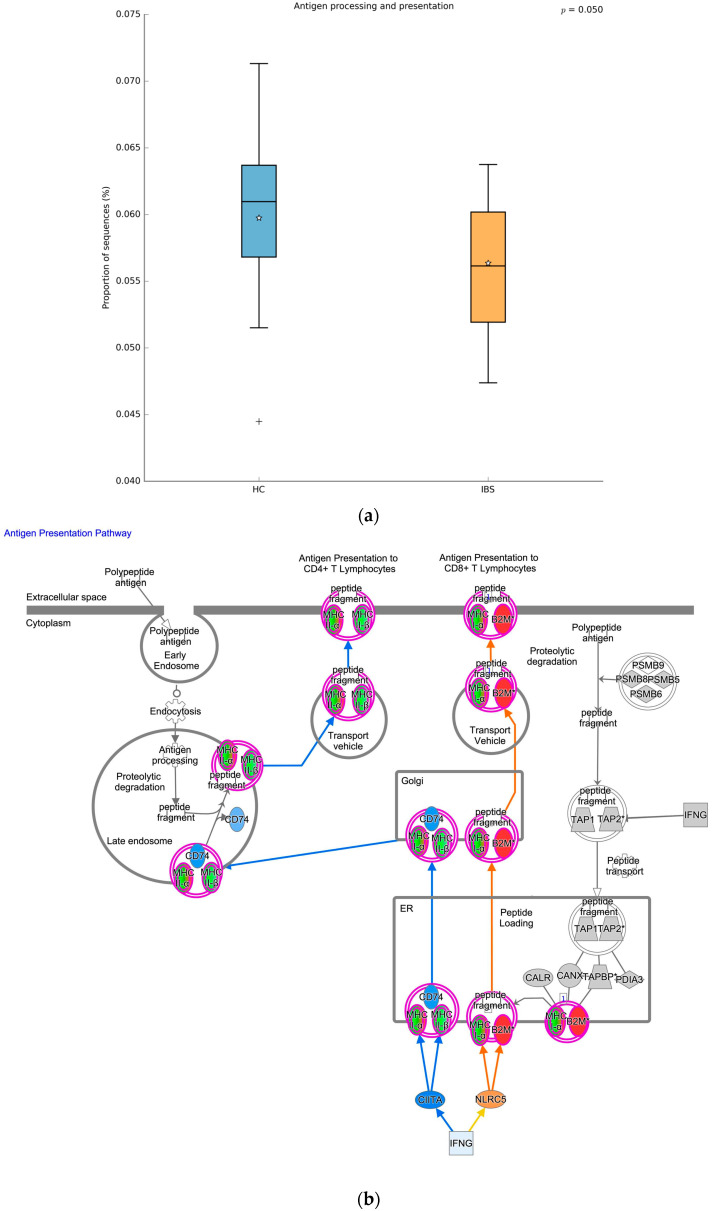
KEGG 04162, antigen processing and presentation in the gut microbiota and DEGs. (**a**). Gut microbiota from stool samples show the predicted KEGG 04162 is significantly enriched in HCs. “Star” indicated the mean value, “+” indicates an outlier. (**b**). Gene expression profiling from blood samples show that the KEGG 04162 is significantly enriched in IBS subjects. Gene, Protein, and Chemical identifiers marked with an asterisk indicate multiple identifiers in the dataset file map to a single gene and/or chemical in the Global Molecular Network.

**Figure 6 ijms-25-03514-f006:**
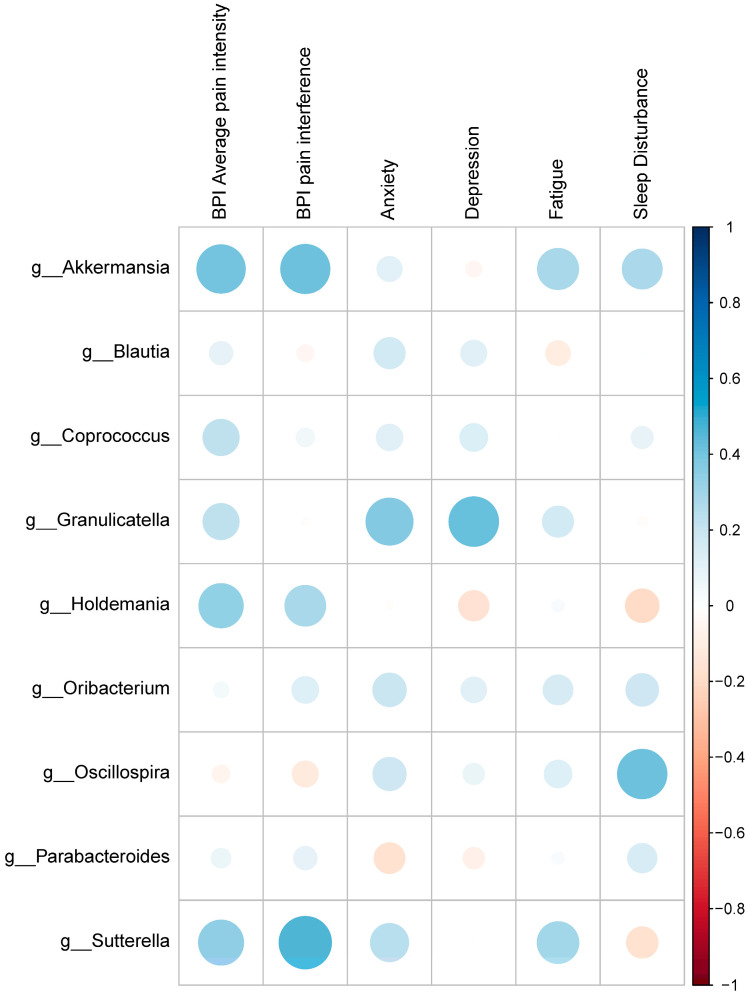
Spearman’s correlation coefficient of the self-reported pain and symptoms and the identified genera. Horizontal axis, self-reported pain and symptoms; and vertical axis, the nine genera significantly enriched in IBS, including *Parabacteroides*, *Akkermansia*, *Blautia*, *Coprococcus*, *Granulicatella*, *Holdemania*, *Oribacterium*, *Oscillospira*, *Parabacteroides*, *Sutterella.* Blue indicates a positive correlation, red indicates a negative correlation. No significant correlations emerged between the enriched 9 genera and self-reported pain and symptoms, after multiplicity adjustment by Benjamini-Hochberg method to control the false discovery rate (FDR) at 0.05.

**Figure 7 ijms-25-03514-f007:**
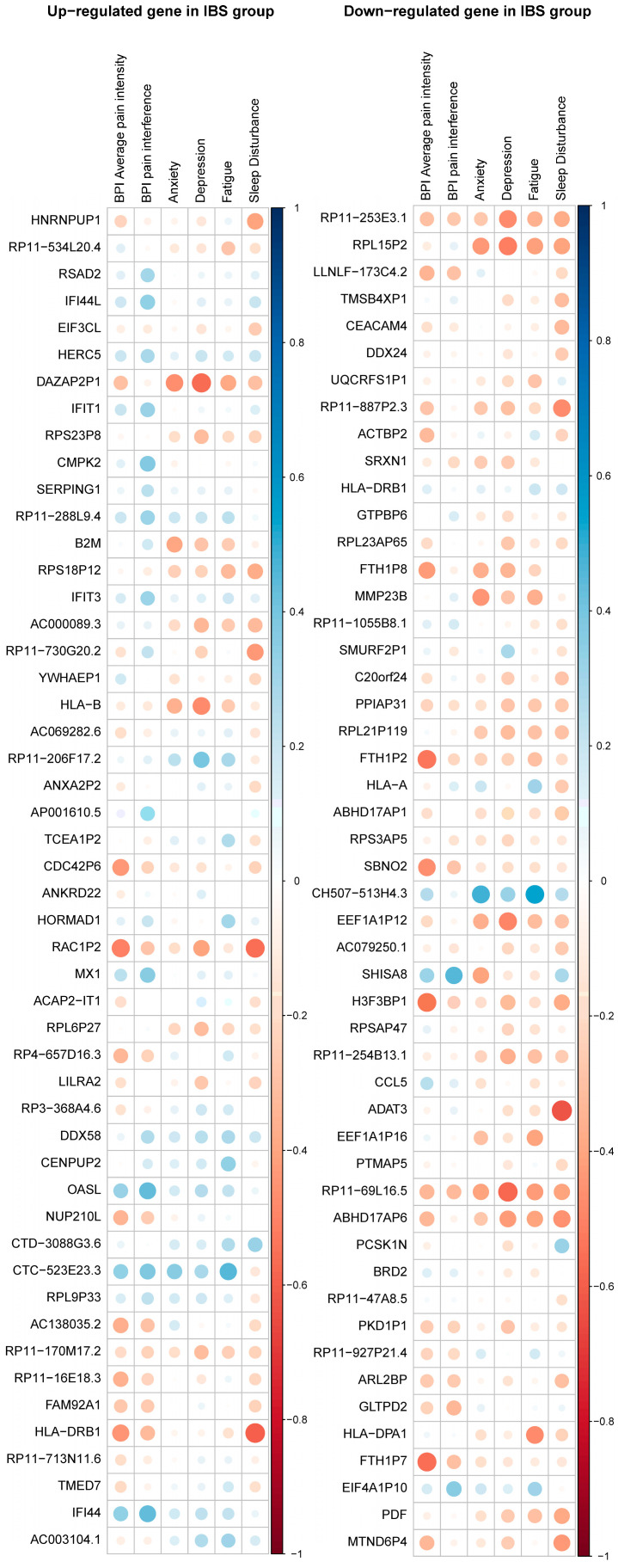
Spearman’s correlation coefficient of the top 50 upregulated and downregulated DEGs and self-reported pain and symptoms. In the left column, the correlations between self-reported pain and symptoms (horizontal axis) and the top 50 upregulated DEGs (vertical axis). In the right column, the correlations between self-reported pain and symptoms (horizontal axis) and the top 50 downregulated DEGs (vertical axis). Blue indicates a positive correlation, red indicates a negative correlation. No significant correlations emerged between the top 50 up- and downregulated DEGs and self-reported pain and symptoms, after multiplicity adjustment by Benjamini-Hochberg method to control the false discovery rate (FDR) at 0.05.

**Table 1 ijms-25-03514-t001:** Demographic characteristics.

	IBS Subjects (N = 20)		Healthy Control (N = 21)		*p*
	n	Percentage (%)	n	Percentage (%)	
Sex					0.530
Female	13	65.00	11	52.38	
Male	7	35.00	10	47.62	
Race					0.144
White	15	75.00	9	42.86	
Asian	2	10.00	6	28.57	
Black or African-American	3	15.00	4	19.05	
Not reported	0	0	2	9.52	
Ethnicity					0.567
Not Hispanic or Latino	15	75.00	16	76.19	
Hispanic or Latino	3	15.00	4	19.05	
Not reported	2	10.00	1	4.76	
Education					0.043
High school or below	0	0	2	9.52	
College or associate degree	11	55.00	17	80.95	
Bachelor degree	5	25.00	2	9.52	
Graduate or Doctorate degree	4	20.00	0	0	
Employment Status					0.094
Student	13	65.00	18	85.71	
Working now	7	35.00	2	9.52	
Unemployed or other	0	0	1	4.76	
Marital Status					0.232
Never married	18	90.00	21	100	
Married	2	10.00	0	0	
	Mean	SD	Mean	SD	
Age	22.05	2.74	20.14	1.39	0.030
Year of IBS diagnosis	2.25	1.89			NA

Note: IBS, irritable bowel syndrome; SD, standard deviation; NA, not applicable. The *p*-values are calculated by Fisher’s exact test for categorical variables and Wilcoxon rank sum test for continuous variable.

**Table 2 ijms-25-03514-t002:** Summary statistics of the pain and symptom variables among the two groups, mean (SD).

	IBS Subjects (N = 20)	Healthy Control (N = 21)	*p*
BPI pain severity	3.05 (2.14)	0.33 (0.72) ^a^	<0.001
Worst pain	4.30 (2.39)	0.62 (1.07)	<0.001
Least pain	2.05 (2.46)	0.10 (0.43)	<0.001
Average pain	3.70 (2.11)	0.43 (0.92)	<0.001
Current pain	2.15 (2.48)	0.19 (0.68)	0.001
BPI pain interference	2.65 (2.77)	0.23 (0.40) ^a^	<0.001
PROMIS Symptoms T-Score			
Anxiety	58.76 (8.30)	54.55 (6.00) ^b^	0.047
Depression	52.53 (8.07)	48.48 (6.12)	0.099
Fatigue	53.53 (9.63)	48.73 (9.21) ^b^	0.042
Sleep disturbance	49.90 (5.84)	47.68 (4.33)	0.375

Note: SD, standard deviation; BPI, brief pain inventory; IBS, irritable bowel syndrome; PROMIS, Patient-Reported Outcomes Measurement Information System. ^a^ The measurement is significantly different from the baseline tested by a Wilcoxon rank sum test; ^b^ the scores for the healthy control group were significantly lower than the IBS groups tested by a Wilcoxon rank sum test.

## Data Availability

The raw sequence data were archived in NCBI (https://submit.ncbi.nlm.nih.gov/subs/sra/SUB8914789/). Deidentified data are available upon reasonable request. Requests to access these datasets should be directed to xiaomei.cong@yale.edu.

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
