# Peer review of "Multi-Omics Analysis of Gut Microbiota and Host Transcriptomics Reveal Dysregulated Immune Response and Metabolism in Young Adults with Irritable Bowel Syndrome"

_ijms, 2024, doi:10.3390/ijms25063514_

Round 1
Reviewer 1 Report
Comments and Suggestions for Authors
The manuscript employs a multi-omics strategy to study the gut microbiota and host transcriptomics simultaneously. This holistic approach is commendable as it offers a comprehensive view of the interactions between the microbiota and host responses. As irritable bowel syndrome (IBS) is a prevalent condition affecting a significant portion of the population, the findings of this study have the potential to impact a wide audience. They contribute to a better understanding of IBS pathophysiology, which is crucial for improving the management of this condition. Overall, the manuscript is well-written and structured, making it accessible to a broad scientific audience. In light of these considerations, I believe the manuscript is suitable for publication after making some appropriate modifications as follows.
The overall research ideas of the article are clear, and the experimental design and research content are good. However, there still some issue need further modification as follow.
Specific comments:
Lines 82 and 86: Delete n = 41. It is the wrong number of repetitions for each group.
The results of PCoA for gut microbiota need be provided.
The results of PCA plot for RNA-seq need be provided.
The bar chart in Figure 1a1-4 requires modification to improve data visualization. Consider adjusting colors, labels, scale, or layout in the chart to make the data clearer to the audience.
Figures: Figure clarity must be enhanced by employing higher-resolution charts and images and ensuring that labels and annotations are legible.
Keywords: Avoid repeating the words used in the title to expand the manuscript's searchability range and make it more easily indexed by academic search engines.
Author Response
We appreciate the Editors and Reviewers very much for their positive, constructive, and valuable suggestions on our manuscript entitled “Multiomics analysis of gut microbiota and host transcriptomics reveal dysregulated immune response and metabolism in young adults with irritable bowel syndrome” (ijms-2874530). We have studied the Editors’ and reviewers’ comments carefully and have made revisions which marked in green (response to Reviewer 1) and in yellow (response to Reviewer 4), respectively in the revised manuscript.
Reviewer 2 Report
Comments and Suggestions for Authors
Despite the small number of subjects, the research presented in this paper is promising and may constitute an interesting introduction to research on a larger population. The study revealed the linkage of gut microbiota and transcriptomic profiles among tested patients and the associated dysregulation of the immune system, which seems to be a probable pathogenesis of IBS. However, studies on a larger group are needed to confirm this with certainty. I propose to accept this quality paper and publish it in the International Journal of Molecular Sciences.
Author Response
We appreciate the Editors and Reviewers very much for their positive, constructive, and valuable suggestions on our manuscript entitled “Multiomics analysis of gut microbiota and host transcriptomics reveal dysregulated immune response and metabolism in young adults with irritable bowel syndrome” (ijms-2874530). We have studied the Editors’ and reviewers’ comments carefully and have made revisions which marked in green (response to Reviewer 1) and in yellow (response to Reviewer 4), respectively in the revised manuscript.
We appreciate the positive feedback.

Reviewer 3 Report
Comments and Suggestions for Authors
General comments:
This study investigated the associations among gut microbiota and host transcriptomics in young adults with irritable bowel syndrome (IBS). The results showed that the feasibility of multi-omics investigations to explore the underlying mechanisms of symptomology in young adults with IBS, as well as in patients with other conditions.
The manuscript could be considered for publication since it was well written.
Author Response

(The authors gave the same response as above.)

Reviewer 4 Report
Comments and Suggestions for Authors
This manuscript is clearly and scholarly written. The authors demonstrated sound scientific knowledge and excellent analysis of the gut microbiota and host transcriptomics data. Their analysis of the data detailed a dysregulated immune response and metabolism, a well-thought-out conclusion.
Multi-omics analysis of gut microbiota and host transcriptomics reveal dysregulated immune response and metabolism in young adults with irritable bowel syndrome by Chen et al., discovered that the pathway of antigen processing and presentation in gut microbiota was different and demonstrated a dysregulated immune response and metabolism in IBS. Their research discovery is a very important piece of scientific evidence that can be deployed in the fight against IBS. I encourage the publication of this work. However, I encourage minor corrections in their manuscript as spelt out below.
Minor Modifications.
1. Figure 1c, has blending colors, e.g. colors for Blautia and Parabacteriodes are almost similar. Please revise.
2. Figures quality in Figure 4 should be improved.
3. Figure 5. “KEGG 04162, Antigen processing and presentation in the gut microbiota and DEGs. Figure 5a. The KEGG 04162 in stool samples was significantly enriched in HCs. Figure 5b. The KEGG 04162 in gene expressions was significantly enriched in IBS.” Please this figure description is confusion. Please revise appropriately.
4. Figure 5. Mhc-2 and MHC-II are the same, please revise appropriately, their differences are the alpha and the beta.
5. Some places, “Figure” is written in bold whereas it is unbold in other places. Please be consistent.
Author Response

(The authors gave the same response as above.)
